# Thermal Spring Waters as an Active Ingredient in Cosmetic Formulations

**Ana Carolina Figueiredo** [1], **Márcio Rodrigues** [1,2], **M. Lourdes Mourelle** [3,*] **and André R. T. S. Araujo** [1,4,*]

1   CPIRN-IPG, Center of Potential and Innovation of Natural Resources, Polytechnic Institute of Guarda, Av. Dr. Francisco de Sá Carneiro, No. 50, 6300-559 Guarda, Portugal
2   CICS-UBI, Centro de Investigação em 3 Ciências da Saúde, Universidade da Beira Interior, Avenida Infante D. Henrique, 6200-506 Covilhã, Portugal
3   FA2 Research Group, Applied Physics Department, University of Vigo, 36310 Vigo, Spain
4   Department of Chemical Sciences, Laboratory of Applied Chemistry, Faculty of Pharmacy, LAQV, REQUIMTE, Porto University, Rua de Jorge Viterbo Ferreira, 228, 4050-313 Porto, Portugal
*   Correspondence: lmourelle@uvigo.es (M.L.M.); andrearaujo@ipg.pt (A.R.T.S.A.)

**Abstract:** Background: Thermal waters have been showing different beneficial effects on the skin due to their physicochemical composition. The beneficial effect of thermal water in the treatment of some skin diseases may thus justify its use as an active ingredient in cosmetic formulations. The main objective of this work was to demonstrate the potential of incorporating thermal water as an active ingredient in cosmetic formulations. (2) Methods: A descriptive literature review was carried out by the analysis of scientific articles in PubMed and Google Scholar databases. Twelve thermal spring waters were found (Avène, Blue Lagoon, Comano, Cró, Dead Sea, La Roche-Posay, Monfortinho, Saint-Gervais, Salies-de-Béarn, São Pedro do Sul, Uriage and Vichy) with potential as an active in cosmetic products, demonstrated through in vitro studies evaluating the different activities/properties and clinical trials in healthy volunteers or with skin pathologies. (3) Results: For these studies, in natura thermal water as well as incorporated in cosmetic formulations were used. In in vitro studies, most thermal waters have been shown to have activities on membrane fluidity, skin barrier repair, antiradical, antioxidant, anti-inflammatory and immunomodulatory properties, proliferative activity, regulation of processes involved in ageing and moisturizing properties. In clinical trials, cosmetic thermal waters reduced skin discomfort through their soothing and exhibited moisturizing and anti-irritant properties. (4) Conclusions: The effect of thermal waters on the skin and the absence of side effects reported in different studies allows them to be used as an adjuvant or in the treatment of various skin disorders and may play an important role in the cosmetics industry. However, further clinical trials are needed to assess their effectiveness and safety.

**Keywords:** thermal water; active ingredient; cosmetic formulations





## 1. Introduction

The skin is considered to be the largest organ in the human body and acts as a protective barrier against physical, chemical or microbiological agents and is responsible for defensive functions, thermoregulatory, metabolic, excretory and sensory [1]. To achieve balance, hydration and the wellbeing of the skin, thermal spring waters are often used [2], these solutions being formed under specific geological conditions and characterized by "physic-chemical dynamism", having three important properties: natural origin, bacterial purity and therapeutic potential [3,4].

The therapeutic effects of thermal spring waters have been shown since ancient times; during Roman empire, the use of these mineralized and frequently hot spring waters was spread throughout Europe, both for therapeutic and recreational use. Additionally, over the last two centuries, thermal spring waters shown different beneficial effects on the skin due to their physicochemical composition.

Thermal spring waters can be classified into different categories, according to their chemical composition and physical elements, such as temperature, molecular concentration and mechanisms of therapeutic action. Taking mineralization into account (dry residue at 110 °C), they can be classified as following: oligometalic (<100 mg/L), very low mineralization (100–250 mg/L), low mineralization (250–500 mg/L), medium mineralization (500–1000 mg/L) and high mineralization (>1000 mg/L) [2]. About the mineral composition, it depends on the country, but the most common types are chloride, bicarbonate, sulfate, sulfide, carbo-dioxide, and weakly mineralized waters. They can also contain soluble minerals as $Mg^{+2}$, $Ca^{+2}$, $Na^+$, silicates, iron compounds, etc., and trace minerals as selenium, zinc, boron, etc. Related to the temperature, there are also many classifications (from the geological or medical point of view); the most widespread is based on the mean annual soil temperature (Ts) or the mean annual air temperature (Ta) as the following: hypothermal (temperature < Ts mean annual + 4 °C), mesothermal (temperature = Ts mean annual + 4 °C), and hyperthermal (temperature > Ts mean annual average + 4 °C). However, for therapeutical purposes, those related to body temperature are most useful: hypothermal (<37 °C), mesothermal (=37 °C), hyperthermal (>37 °C).

The thermal spring waters used to treat dermatologic conditions present different identities in terms of physicochemical profile [3,4]. Most of them are sulfur, sulphate or bicarbonate thermal waters; others are chloride or rich in silica, calcium, magnesium, etc., or in trace elements as zinc, selenium, boron, etc. [2].

The mechanisms by which thermal water act, in spa therapy, involve chemical, thermal, mechanical and immunological effects on different dermatologic conditions, such as atopic dermatitis (AD), contact dermatitis, seborrhea, seborrheic dermatitis, psoriasis and ichthyoses [3–5].

Cosmetic formulations are "any substance or mixture intended to be placed in contact with the external parts of the human body (epidermis, hair system, nails, lips and external genital organs) or with the teeth and the mucous membranes of the oral cavity with a view exclusively or mainly to cleaning them, perfuming them, changing their appearance, protecting them, keeping them in good condition or correcting body odors" (EU Regulation 1223/2009, Article 2.1.a). Cosmetics are made of several active ingredients and an excipient; water is the main excipient in most formulas (lotions, creams, etc.); thus, mineral/thermal spring water could be used both as active ingredient or as a part of the excipient.

In recent decades, the cosmetic industry has discovered increasing knowledge of normal skin physiology as well as the development of new research techniques and consequently advances in knowledge, and of novel active ingredients and vehicles, based on well-understood mechanisms of action [6].

Considering the beneficial effects of thermal spring waters in the treatment of some skin diseases, and since they are rich in minerals and trace elements with proven dermatological indications, they may thus justify their use as an active ingredient in cosmetic formulations. The main objective of this work was to demonstrate the potential of incorporating thermal spring waters as an active ingredient in cosmetic formulations, providing cosmetics with its minerals and trace elements.

## 2. Materials and Methods

For the nonsystematic literature review, PubMed and Google Scholar databases were consulted, and the search terms used were Thermal Water, Dead Sea and Cosmetic Formulations.

To ensure that relevant information was obtained to answer the research question, inclusion and exclusion criteria were established. The inclusion criteria were articles focusing on the problem under study, articles describing the composition of the thermal water under study, articles describing the different specific activities of thermal water and articles reporting the therapeutic use of thermal water. As an exclusion criterion, articles not related to the problem under study and in which thermal water was not one of the active ingredients of the formulation were considered. Thus, 50 articles were analyzed for review.

Data were extracted and presented in a table for a more accurate presentation of the results. The table is organized in alphabetical order of thermal waters and contains the origin of each water, its constituents, from the majority to the minority, the cosmetic formulations used in the studies or already marketed, the types of studies analyzed, the effects observed in these studies, the activities specific of each thermal water, as well as the therapeutic uses resulting from these activities.

## 3. Results

*Subsection*

From the analysis of scientific articles, 12 thermal spring waters were found (Avène, Blue Lagoon, Comano, Cró, Dead Sea, La Roche-Posay, Monfortinho, Saint-Gervais, Salies-de-Béarn, São Pedro do Sul, Uriage and Vichy) with potential as an active ingredient in cosmetic products, demonstrated through in vitro studies evaluating the different activities/properties and clinical trials in healthy volunteers or with skin pathologies.

The thermal waters used in the treatment of dermatological conditions present different identities in terms of physicochemical profile, each having specific activities (Table 1).

**Table 1.** Main results of the studies involving thermal spring waters.

| Thermal Water | Origin | Composition | Cosmetic Formulations | Type of Study | Observed Effects | Specific Activities | Therapeutic Use | References |
|---|---|---|---|---|---|---|---|---|
| Avéne | Lavour (France) | Bicarbonate Calcium Magnesium Silica Sulfate Chloride Potassium Sodium | Spray | In vitro studies | Inhibition of mast cells activation Inhibition of degranulation induced by substance P Inhibition of the TNF-$\alpha$-induced E-selectin and ICAM-1 expression Suppression of NF-$\kappa$B transcription factor pathway activation Reverse the induction of IL-6 and the formation of ROS after UVB stimulation on human keratinocyte HaCaT cells | Effect on membrane fluidity Antiradical properties Anti-inflammatory effects Immunomodulatory effects Cell differentiation Antioxidant properties | Acne Atopic dermatitis | [7–13] |
| | | | | Clinical trials | Decreased erythema Pain reduction Pruritus reduction | Anti-irritant activity Soothing properties | Actinic keratosis Acne Post photodynamic therapy Dermal melasma | [14,15] |
| | | | | Clinical and inflammatory biomarkers study | Clinical symptoms reduction Reduction of the potentially pathogenic bacteria in growth phase | Anti-inflammatory properties Preventing growth bacteria | Psoriasis Atopic dermatitis | [16] |
| | | | | Ex vivo study and clinical study | Maintaining mechanical properties and hydration after a chemical peeling Redness reduction Skin sensitivity reduction | Protection against dehydration Anti-irritant activity | Post-chemical peeling intervention | [17] |

Table 1. *Cont.*

| Thermal Water | Origin | Composition | Cosmetic Formulations | Type of Study | Observed Effects | Specific Activities | Therapeutic Use | References |
|---|---|---|---|---|---|---|---|---|
| | | | Emollient cream containing an *Aquaphilus dolomiae* extract | Open-label, real-world study | Significant improvements in xerosis and pruritus severity Reduction in itch duration SCORAD improvement Improvements in sleep quality and DLQI scores Reduction of xerosis severity in patients treated for cancer | Soothing and skin hydration properties Anti-irritant activity | Xerosis pruritus | [18,19] |
| | | | | | Significant improvements in xerosis and pruritus severity Reduction in itch duration SCORAD improvement Improvements in sleep quality and DLQI scores Reduction of xerosis severity in patients treated for cancer | | | |
| Blue Lagoon | Grindavik (Iceland) | Chloride Sodium Potassium Calcium Silica | Natural thermal water combined with NB-UV + moisturizing cream | Clinical and ex vivo studies | SCORAD improvement Reduction in circulating CLA$^+$ peripheral blood T cells Decrease of Th1/Th17 and Tc1/Tc17 inflammatory response | Anti-inflammatory properties | Psoriasis | [20,21] |
| | | | Extracts from microalgae and silica mud Cream made of microalgae and silica mud extracts | In vitro and in vivo studies | Silica mud extracts and coccoid and filamentous algae extracts: induction the expression of genes relevant for keratinocyte differentiation such as transglutaminase 1, filaggrin, and involucrin Coccoid and filamentous algae extracts: increase collagen gene expression Cream: increase mRNA expression for involucrin, filaggrin and transglutaminase-1 and induction collagen 1A1 and 1A2 mRNA expression; UV-induced gene expression reduction; decrease in TEWL | Skin barrier improvement Protection against extrinsic skin ageing | Healthy skin UVA irradiated skin | [22] |

**Table 1.** *Cont.*

| Thermal Water | Origin | Composition | Cosmetic Formulations | Type of Study | Observed Effects | Specific Activities | Therapeutic Use | References |
|---|---|---|---|---|---|---|---|---|
| | | | Exopolysaccharides from *Cyanobacterium aponinum* | In vitro studies | Increase IL-10 secretion by human dendritic cells<br>Increase differentiation of T cells into T regulatory cells<br>Attenuate T cell activation evidenced by lowered proportion of the cells expressing CD69 and a decrease in their cytokine secretion<br>Reduce secretion of the chemokines CXCL10 and CCL20<br>Reduced inflammatory cell recruitment<br>Reduce keratinocyte production of LL37 inactivation of the Dectin-1 receptor | Anti-inflammatory properties | Psoriasis | [23,24] |
| | | | Blue Lagoon algae extracts Cream composed of Blue Lagoon algae extracts | In vitro and in vivo studies | Decrease of the expression of α-melanocyte-stimulating hormone-induced expression of genes involved in melanin synthesis<br>Reduction of number of pigmentation spots | Uneven skin pigmentation | | [25] |
| Comano | Comano-Trentino (Italy) | Bicarbonate Calcium Sulphate Magnesium | Natural thermal spring water | In vitro studies | Reduction of all vascular endothelial growth factor-A-mediated angiogenic, vessel permeabilizing, and chemotactic effects<br>Reduction of intracellular levels and secretion rates of IL-6<br>Downregulation of the expression of cytokeratin-16<br>Improvement of cell vitality of the human keratinocyte's cultures | Reduction of abnormal differentiation | Psoriasis Wound healing | [26–29] |
| | | | | In vivo experimental study | Increase keratinocyte proliferation and migration<br>Modulation of the regenerated collagen and elastic fibers in the dermis | Improvement of skin regeneration | Wound healing | [30] |
| | | | | Ex vivo model | Markable anti-inflammatory effect by reducing overall dermal cell infiltration | Tissue regeneration and wound healing | Wound healing | [31] |



**Table 1.** *Cont.*

| Thermal Water | Origin | Composition | Cosmetic Formulations | Type of Study | Observed Effects | Specific Activities | Therapeutic Use | References |
|---|---|---|---|---|---|---|---|---|
| Cró | Beira Interior (Portugal) | Bicarbonate Sodium Silica Calcium Potassium Magnesium | Gel | In vitro study and clinical trial | Promotes the normal human dermal fibroblasts adhesion and proliferation Hydration increase Decrease in TEWL Lesser roughness Lower scaliness Higher smoothness Skin relief improvement | Cell proliferation Hydration properties | - | [32] |
| Dead Sea | Dead Sea (Israel) | Magnesium Calcium Potassium Sodium Strontium Chloride Bromo | Natural thermal spring water and anionic polysaccharide (PolluStop®) | In vitro study | Inhibition of IL-1α and prostaglandin E2 overproduction | Anti-inflammatory properties | Antipollution skin protection | [33] |
| | | | Cream (Dermud™) | In vitro study | Reversal of decrease of mitochondrial activity and increase of caspase 3 activity after UVB exposure application Inhibition of the secretion of TNF-α and IL-1α, IL-6 and IL-8 | Protective, antioxidant and anti-inflammatory properties | - | [34] |
| | | | Cream | Clinical trial | Improvement of OSAAD score, TEWL, stratum corneum hydration | - | Atopic dermatitis | [35] |
| La Roche-Posay | La Roche-Posay (France) | Bicarbonate Calcium Silica Magnesium Strontium Selenium | Natural mineral spring water | In vitro studies | Better cell survival Reduced IL-1α, IL-6, TNF-α release Reverse the induction of IL-6 and the formation of ROS after UVB stimulation on human keratinocyte HaCaT cells Decrease migration Langerhans cells Decrease IL-6 production, both at the intracellular and extracellular levels Increase selenium-dependent glutathione peroxidase activity Decrease lipoperoxides production | Radical scavenger properties Anti-inflammatory properties Immunomodulatory properties | - | [11,36–40] |

Table 1. *Cont.*

| Thermal Water | Origin | Composition | Cosmetic Formulations | Type of Study | Observed Effects | Specific Activities | Therapeutic Use | References |
|---|---|---|---|---|---|---|---|---|
| | | | | Clinical trials | Reduce number of sunburn cells Reduced redness and telangiectasia intensity after 1 month treatment | Protection against UVB Anti-inflammatory properties Anticarcinogenic properties | Healthy skin Rosacea | [41,42] |
| | | | Gel | In vivo study | Blood flow reduced after sodium lauryl sulphate irritation | Anti-inflammatory effect | Healthy skin | [42] |
| | | | Cream | In vivo study | Glutathione peroxidase activity increased | Protection against UVB | - | [43] |
| Monfo-rtinho | Idanha à Nova (Portugal) | Bicarbonate Silicate Sodium Magnesium Calcium Potassium | Cream | Clinical trial | Improvement of erythema Decreased pruritus | Skin hydration | Psoriasis Eczema | [44] |
| | | | | In vitro study | Reduction on cell metabolism and proliferation | Antiproliferative effect Anti-inflammatory properties | Atopic dermatitis Psoriasis | [45] |
| São Pedro of Sul | São Pedro do Sul (Portugal) | Bicarbonate Sodium Silica Chloride Fluoride Silicate Sulphate | Natural mineral spring water | Clinical trial | Decreased TEWL | Anti-irritant effect | Skin irritation | [46] |
| Saint-Gervais Mont Blanc | Saint-Gervais les Bains (France) | Sulphate Bicarbonate Sodium Calcium Manganese Boron | Natural mineral spring water | In vitro study | Promotion migration of keratinocytes Improvement of barrier function | Wound healing | Scars | [47] |
| Salies-de-Béarn | Béarn des Gaves (France) | Bicarbonate Calcium Magnesium | Natural mineral spring water | Clinical trial | Decreased PASI | Minor therapeutic effects in psoriasis | Psoriasis | [48] |

| Thermal Water | Origin | Composition | Cosmetic Formulations | Type of Study | Observed Effects | Specific Activities | Therapeutic Use | References |
|---|---|---|---|---|---|---|---|---|
| Uriage | Alpes (France) | Sulphate Chloride Sodium Bicarbonate Calcium Magnesium | Natural mineral spring water | In vitro study | Effect on taurine transporter and sodium-dependent vitamin C transporter 1 expression | Regulation of the processes involved in aging | Skin fight against stressful situations such as dehydration, UVB irradiation and aging | [49] |
| | | | Cream | In vitro studies | Increased expression of claudin-4, claudin-6, filaggrin and aquaporine-3 | Skin hydration | Dry skin | [50,51] |
| | | | | In vitro study | Increase of human dermal fibroblasts Reduce the lipid peroxidation through thiobarbituric acid reactive substances assay Recovery of catalase activity after UV irradiation Restoration of claudin-6 expression after UVB irradiation | Antioxidant Properties DNA protection | DNA protection of the cutaneous tissue in front of the UV irradiations | [50] |
| | | | UTSW + rhamnose-rich polysaccharide (PS291®) | In vitro study | Increase of the generation time and reduction of biomass of *Cutibacterium acnes* (strain RT4 and RT5 acneic) Reduction of final biomass of *Staphylococcus aureus* | Antibiofilm activity | - | [52] |
| | | | Natural mineral spring water | In vitro study | Counteract the increase of biofilm formation of RT4 acneic strain of *C. acnes* after exposure to epinephrine; similar result with norepinephrine, but UTSW could not completely inhibit the effect of norepinephrine Decrease in RT6 biofilm formation but an exposure of the bacterium to epinephrine in the presence of UTSW induced a limited but significant increase in the biofilm | Antibiofilm activity in the presence of catecholamines | - | [53] |

Table 1. *Cont.*

| Thermal Water | Origin | Composition | Cosmetic Formulations | Type of Study | Observed Effects | Specific Activities | Therapeutic Use | References |
|---|---|---|---|---|---|---|---|---|
| Vichy | Auvergne (France) | Magnesium Potassium Calcium Sulphate Sodium | Natural mineral spring water | In vitro study | Increased expression of genes related to cutaneous homeostasis | Cell proliferation–differentiation balance role in hydration Antioxidant mechanisms and DNA repair | Skin ageing exposome | [54] |
| | | | Natural mineral spring water | In vitro study | Increased transglutaminase, filaggrin, involucrin, claudin-1, and zonula occludens-1 Increased the expression of β-defensin-4A and S100A7 Down-regulated IL-8, TNF-α, IL-12/IL-23p40, and increased IL-10 and IL-10/IL-12 Protected Langerhans cells in skin explants exposed to UV radiation | Skin barrier function Antimicrobial peptide defenses Immune defense functions Protection of Langerhans cells challenged by UV radiation | Strengthen the skin barrier function | [55] |
| | | | Dermocosmetic formulation: Minéral 89 Probiotic Fractions (M89PF) | Clinical trial | Improved skin renewal Better microbiome recovery after acute stress from a harsh cleanser Depigmenting properties on dark spots | Skin barrier effects Skin antioxidant defense activity Depigmenting properties | Prevent and repair skin barrier disruption and reinforce skin defenses in skin exposed to acute stresses | [56] |

IL: interleukin; OSAAD: objective severity assessment of atopic dermatitis; ICAM-1: intercellular adhesion molecule 1; NF-κB: nuclear factor kappa B; PASI: psoriasis area severity index; SCORAD: SCORing atopic dermatitis; ROS: reactive oxygen species; UTSW: Uriage thermal spring water; UV: ultraviolet; TEWL: transepidermal water loss; TNF-α: tumor necrosis factor alfa.

## 4. Discussion

*4.1. Avène Thermal Spring Water*

Avène thermal spring water (ATSW) is characterized by low mineral content. This thermal spring water also contains bicarbonate and presents a stable ratio of calcium and magnesium content (2/1) (Table 2) [11,14].

**Table 2.** Avène thermal spring water composition (adapted from [11]).

| Anions/Cations/Trace Elements and Other Compounds | |
|---|---|
| Bicarbonate (mg/L) | 226.7 |
| Sulphate (mg/L) | 13.1 |
| Chloride (mg/L) | 5.4 |
| Nitrate (mg/L) | 1.4 |
| Fluoride (mg/L) | 0.1 |
| Phosphate (mg/L) | 0.3 |
| Silica $SiO_2$ (mg/L) | 14 |
| Calcium (mg/L) | 42.7 |
| Magnesium (mg/L) | 21.2 |
| Potassium (mg/L) | 0.8 |
| Sodium (mg/L) | 4.8 |
| Iron (mg/L) | <0.1 |
| Manganese (mg/L) | <0.1 |
| Strontium (mg/L) | 0.1 |
| Lithium (mg/L) | <0.1 |
| Boron (µg/L) | 220 |
| Cadmium (µg/L) | 20 |
| Zinc (µg/L) | 2 |
| Copper (µg/L) | <5 |
| Selenium (µg/L) | <5 |
| Barium (µg/L) | 220 |

Avène dermocosmetics are very well-known around the world, being sold in pharmacies and other cosmetic sale centers mainly in Europe but also in other continents and countries, including Japan. Over the last twenty years, many studies have been carried out to demonstrate its effectiveness.

Two clinical trials evaluated the use of ATSW in terms of efficacy and tolerance when used as an adjunctive care in post procedure skin care with photodynamic therapy. In the comparative's studies, twenty-five patients suffering from either vulgaris acne or photodamage with or without actinic keratoses and twenty patients with bilateral dermal melasma were included. The first study compared the effects of ATSW low mineral content spring water to a high mineral content spring water (both applied four times a day along one week). In the second, ATSW was sprayed in half-face along two days, six times a day. In both studies, ATSW reduced erythema and pruritis and relieved pain. The studies conducted showed that spraying ATSW after photodynamic therapy significantly reduced short-term adverse effects associated with the procedure. The results obtained showed the smoothing and anti-irritant properties ATSW due to the involvement in the reduction in sensitivity of cells and the anti-inflammatory effects [14,15].

Several in vitro studies were performed during the last twenty years, mainly in AD. Joly et al. [7] studied two different thermal spring waters ("Sainte Odile" and "Val d'Orb") from Avène on rat peritoneal mast cell activation. In this study, cells were preincubated with both Avène spring waters. A dilution-dependent inhibition of both histamine and prostaglandin $D_2$ antigen-induced release was observed, and also inhibited histamine release triggered by substance P. The authors suggested that the ability of ATSW to inhibit mast cell activation in vitro may be related to its antiallergic and anti-inflammatory properties.

Charveron et al. [13] investigated, in normal keratinocytes irradiated with ultraviolet-A (UVA) (365 nm, 31 J/cm$^2$), the antioxidant effect; these in vitro experiments demonstrated

that ATSW was capable of protecting cell membranes, genomic DNA and proteins from the UVA-induced oxidative stress. Additionally, other investigation observed the effect of ATSW on the release of histamines by mast cells. A significant inhibitory effect on the Substance P- or antigen-induced cells degranulation (histamine or prostaglandin D-2 release) was observed after ATSW treatment [8].

Other studies aimed to assess the anti-inflammatory effects. In a model of human skin explants stimulated by a neurotransmitter (vasoactive intestinal peptide), ATSW showed a significant reduction in the inflammatory effect [9].

An in vitro study evaluated the beneficial effect of ATSW on AD. The study consisted of treating a set of human endothelial cells with tumor necrosis factor-$\alpha$ (TNF-$\alpha$), which stimulates reactive oxygen species (ROS) that can function as second messengers in the production of E-selectin expression, in the presence or not of ATSW. It was observed that the treatment with ATSW significantly inhibited TNF-$\alpha$-induced expression of E-selectin and intercellular adhesion molecule 1 (ICAM-1) by 22% and 7%, respectively, and inhibited the activation and translocation of transcription factor nuclear factor kappa B (NF-$\kappa$B) to the nucleus in cells treated with TNF-$\alpha$, possibly due to its antioxidant properties. Through the inhibition of adhesion molecules involving NF-$\kappa$B, the ATSW exhibited a regulation of inflammatory parameters, concluding that this water can minimize the inflammatory reaction in AD [10].

In addition, one study focused on the effect of ATSW on the skin barrier. After incubation of NHK cells in the presence of a medium reformed with ATSW, an overexpression of early differentiation markers such as Keratin K1 and K10 or late differentiation markers (involucrin, transglutaminase 1, filaggrin or ABCA12) was shown [12].

Finally, it should be cited that the study conducted by Zöller et al. [11], where they compared the effect of culture media supplemented with (a) thermal spa waters (La Roche-Posay, Avène) and (b) two natural mineral drinking waters (Heppinger, Adelholzener) on physiological parameters in HaCaT keratinocytes, concluding that, despite having different compositions, they both efficiently suppressed the induction of a prototypical inflammatory cytokine (interleukin (IL)-6) and the formation of ROS after UVB stimulation [11].

A clinical and inflammatory biomarker study was performed by Casas et al. [16] with two group of patients (suffering psoriasis and AD) that were treated with several therapeutical techniques, including spraying with ATSW. The decrease both in SCORing atopic dermatitis (SCORAD) and psoriasis area severity index (PASI) was associated with a significant reduction of IL-8 and *Staphylococcus aureus* colonization.

On the other hand, a recent ex vivo and clinical study conducted by Mias et al. [17] showed that ATSW is able to decrease redness and reduce the overall sensitive scale after a dermatological chemical peeling.

There are also studies about the effectiveness of a cream containing an *Aquaphilus dolomiae* extract (a cyanobacteria found in ATSW aquifer). In open-label, real-world studies, this cream was able to reduce pruritus and xerosis in a range of dermatologic and systemic diseases [18]. Similarly, a cream made of the same extract was effective in reducing xerosis in cancer patients, regardless of the initial grade of xerosis and the anticancer treatment received [19].

*4.2. Blue Lagoon Thermal Water*

The Blue Lagoon in Iceland is a geothermal lagoon containing a mixture of seawater and freshwater that formed when warm saline fluid was discharged onto a lava field after a geothermal power plant was built in the area in 1976 [21]. It is a highly mineralized water, with an extremely high concentration of silica (Table 3). In 1993, a laboratory was built, and the development of Blue Lagoon skincare started. Later on, assembling a network of experts, the Blue Lagoon Research and Development Center was created near the Lagoon.

**Table 3.** Blue Lagoon thermal water composition (adapted from [21]).

| Anions/Cations/Trace Elements and Other Compounds | |
|---|---|
| Silica $SiO_2$ (mg/L) | 137 |
| Sodium (mg/L) | 9280 |
| Potassium (mg/L) | 1560 |
| Calcium (mg/L) | 1450 |
| Magnesium (mg/L) | 1.41 |
| $CO_2$ (mg/L) | 16.5 |
| Sulphate (mg/L) | 38.6 |
| Chloride (mg/L) | 18,500 |
| Fluoride (mg/L) | 0.14 |

Since 1994, several studies have been carried out, focused on the therapeutic effect of this silicious water, mainly in psoriatic patients [20,21,57–59]. They also studied the microbial diversity, finding that the dominant microorganisms in the water are *Silicibacter lacuscaerulensis* and cyanobacteria, not found under similar conditions anywhere else in the world [21,60]; later on, the effects of bioactive molecules of this microalga were investigated [21–23,25].

In a clinical and ex vivo study, Eysteinsdóttir et al. [20] compared two groups of patients who received either receiving either thermal water bath combined with narrow-band ultraviolet B (NB-UVB) or only NB-UVB treatment, and moisturizing cream after the treatment; the control group were healthy patients. They evaluated PASI score and determined serum inflammatory cytokines, and skin biopsies were also performed in order to evaluate the severity of the disease using Trozak's histological grading score. The results showed that both treatment regimens demonstrated significant clinical improvements. Moreover, the data suggested that patients receiving combined treatment demonstrated better clinical response, assessed by the PASI score, than patients treated only with NB-UVB. In the immunological evaluation, the study showed that compared with healthy controls, psoriasis patients with active disease had significantly higher proportion of peripheral $CLA^+$ T cells expressing CCR10 and CD103 and T cells with both Th1/Tc1 ($CD4^+$/$CD8^+$ IFN-$\gamma^+$ or TNF-$\alpha^+$ cells) and Th17/Tc17 ($CD4^+CD45R0+IL$- $23R^+$, $CD4^+$/$CD8^+$ IL-$17A^+$ or IL-$22^+$ cells) phenotypes. Both treatments showed a significant clinical improvement effect; nevertheless, bathing in geothermal seawater combined with NB-UVB therapy was more effective compared with NB-UVB monotherapy. This clinical improvement was reflected by a reduction in peripheral blood $CLA^+$ T cells and by a decreased Th1/Th17 and Tc1/Tc17 inflammatory response. The authors concluded that these findings suggest that the inflammatory response in psoriasis is predominantly driven by both $CD4^+$ and $CD8^+$ T cells of the Th17/Tc17 lineages.

Further clinical studies conducted by the same research group, combining bathing in geothermal water and NB-UVB vs NB-UVB monotherapy in psoriatic patients, concluded that bathing in geothermal seawater combined with NB-UVB therapy in psoriasis induces faster clinical (quality of life) and histological score improvement, causes longer remission time and allows lower NB-UVB doses than UVB therapy alone [21].

Several studies related to the effects of the bioactive compounds obtained from Blue Lagoon geothermal microalgae and silica mud. In in vitro and in vivo studies, Grether-Beck et al. [22] investigated the extracts from silica mud and two microalgae (coccoid algae and filamentous algae). Firstly, they studied the human epidermal keratinocyte and human dermal fibroblast viability treated with both extracts and they concluded that, in general, keratinocytes tolerated all treatments better than fibroblasts, the streamed silica mud did not reduce the viability of both cell types over a concentration ranging up to 5000 µg/mL, whereas the algae extracts were more toxic at higher concentrations. They also evaluated the genes involved in keratinocyte differentiation by measuring the transcriptional expression of involucrin, filaggrin and transglutaminase-1 as surrogate markers for keratinocyte differentiation. The results showed that stimulation of keratinocytes with silica mud ex-

tracts increased mRNA steady-state levels for involucrin, filaggrin and transglutaminase-1 in a time- and dose-dependent manner. Similarly, expression of keratinocyte differentiation markers was also increased upon stimulation of cells with extracts from algae coccoid type, although to a lesser extent from the algae filamentous type. The study also showed that, in relationship with collagen gene expression in human dermal fibroblasts, both collagen 1A1 and 1A2 were significantly upregulated in cells treated with extracts from Blue Lagoon coccoid and filamentous algae and to a lesser extent in cells treated with silica mud.

To assess the relevance of these previous findings, they conducted an in vivo study in which they analyzed the identical parameters in healthy human skin of 20 volunteers that had been treated with a galenic formulation containing all three extracts studied. After topical application of this preparation once daily for total of 4 weeks significantly increased mRNA expression for involucrin, filaggrin and transglutaminase-1 were found. In addition, topical application of the Blue Lagoon extracts also induced collagen 1A1 and 1A2 mRNA expression in unirradiated skin after 4 weeks of treatment. Following that buttock skin was exposed to a single dose of UVA radiation, significant upregulation of matrix metalloproteinase-1, IL-1 and IL-6 mRNA expression was observed in untreated skin areas. In marked contrast, UV-induced gene expression was significantly reduced in the contralateral skin sites that had been treated with the Blue Lagoon extracts prior to UV exposure. The authors concluded that these studies have shown that silica mud and these two algae species from the Blue Lagoon contain biologically active material that can be used for skin barrier improvement and protection against extrinsic skin ageing, and further studies are necessary to identify the specific biological compounds that are involved in these effects [22].

Gudmundsdottir et al. [24] conducted two in vitro studies to assess the effects of the exopolysaccharides (EPSs) secreted by *Cyanobacterium aponinum* (EPS-Ca), a prevalent organism in the Blue Lagoon. In the first study, they matured human monocyte-derived dendritic cells in the absence or presence of EPS-Ca and measured the secretion of cytokines by ELISA and the expression of surface molecules by flow cytometry. The results showed that that EPSs secreted by *C. aponinum* stimulated dendritic cells to produce vast amounts of the immunosuppressive cytokine IL-10. These dendritic cells induced differentiation of allogeneic CD4$^+$ T cells with an increased T regulatory but decreased Th17 phenotype. These authors hypothesized that EPSs from *C. aponinum* would contribute to the psoriasis treatment benefits associated with Blue Lagoon bathing. In the second study, using dendritic cells, T Cells and normal adult human primary epidermal keratinocytes, the effect of EPS-Ca in cell activation, cytokine´s and chemokine´s secretion, and expression of some keratinocyte genes was studied. EPS-Ca increased the proportion of dendritic cells expressing CD141, and decreased T cell secretion of IL-17, IL-13 and IL-10. In addition, EPS-Ca reduced keratinocyte secretion of chemokines CCL20 and CXCL10 that are involved in recruitment of inflammatory cells. EPS-Ca decreased DC expression of Dectin-1/CLEC7A and SYK, keratinocyte expression of CLEC7A, SYK and CAMP (the gene for antimicrobial peptide LL37), and T cell expression of phosphorylated Zap70. These results indicated that EPS-Ca may induce a regulatory phenotype of dendritic cells, T cells that are less active/inflammatory and less prone to being retained in the skin, and keratinocytes that induce less recruitment of inflammatory cells to the skin and that these effects may be mediated by the effects of EPS-Ca on CLEC7A and SYK. The authors concluded indicating that EPS-Ca may be beneficial in psoriatic patients [23].

On the other hand, Grether-Beck et al. [25] investigated the effects of the Blue Lagoon algae extracts on skin pigmentation. They developed an in vitro study, investigating the gene expression of epidermal melanocytes, followed by an in vivo study with volunteers suffering from skin pigmentation disorders. In the in vitro study, they analyzed the α-melanocyte-stimulating hormone-induced expression of mRNAs in epidermal melanocytes treated with Blue Lagoon algae extracts. The results showed a significantly reduction of α-melanocyte-stimulating hormone-induced expression of genes essential for melanin synthesis, such as tyrosinase, tyrosinase-related protein 1, dopachrome tautomerase, melan

A protein and pre-melanosome protein. In an in vivo study, 60 volunteers with pre-existing facial pigment spots were treated twice daily with a Blue Lagoon algae containing serum or a vehicle control. The results showed that constitutive skin pigmentation, determined by colorimetry (individual typology angle and luminescence), did not differ significantly between both treatments (serum vs vehicle). In marked contrast, digital photography under cross-polarized lighting and RBX technology (VISIA CR) revealed that the number of pigment spots in the serum-treated face decreased significantly compared to the vehicle-treated side. Thus, extracts of blue lagoon algae were proposed as a potential treatment for uneven skin pigmentation.

*4.3. Comano Thermal Spring Water*

Comano thermal spring water (CTSW), located in northern Italy, is an oligometalic rich in bicarbonate, calcium and magnesium ions (Table 4) [61].

**Table 4.** Comano thermal spring water composition (adapted from [61]).

| Anions/Cations/Trace Elements and Other Compounds | |
|---|---|
| Bicarbonate (mg/L) | 196.56 |
| Sulphate (mg/L) | 6.9 |
| Chloride (mg/L) | 0.80 |
| Nitrate (mg/L) | 0.30 |
| Fluoride (mg/L) | 0.43 |
| Phosphate (mg/L) | 0.03 |
| Silica $SiO_2$ (mg/L) | 4.90 |
| Calcium (mg/L) | 48.90 |
| Magnesium (mg/L) | 12.16 |
| Potassium (mg/L) | 0.5 |
| Sodium (mg/L) | 2.0 |
| Iron (mg/L) | 0.010 |
| Manganese (mg/L) | 0.0016 |
| Strontium (mg/L) | 0.23 |
| Zinc (μL) | 0.043 |
| Copper (μg/L) | 0.005 |

CTSW is used as an active ingredient to prepare several cosmetic formulations focus on sensitive skin, dermatitis and psoriasis. In advertising, they claim that "its documented anti-inflammatory, soothing and alleviating properties keep the skin hydrated, fresh and supple". This statement is supported by various clinical and in vitro studies (Table 1).

Most of the clinical studies are referred to balneotherapy, showing that CTSW has beneficial therapeutic effects on patients with psoriasis [61], eczematous dermatitis [62] and AD [63].

Two in vitro studies were performed by Chiarini et al. [28] with the aim of investigating the effect of CTSW in psoriatic epidermal keratinocytes. In the first assay, the results showed the CTSW was able to interfere with vascular endothelial growth factor-A isoform expression and secretion by the psoriatic keratinocytes. These effects would reduce all vascular endothelial growth factor-A-mediated angiogenic, vessel permeabilizing, and chemotactic effects; thereby, authors suggested that least in part explains the beneficial actions of CTSW balneotherapy on the clinical manifestations of psoriasis. In the second, researchers used IL-6-hypersecreting keratinocytes isolated from six psoriatic patients and found that intracellular levels and secretion rates of IL-6 were drastically curtailed when cells have been cultivated with CTSW. Moreover, CTSW exposure also promptly, intensely and persistently downregulated the expression of cytokeratin-16, a marker associated with the keratinocyte psoriatic phenotype. Finally, the authors concluded that CTSW balneotherapy may beneficially affect the clinical manifestations of psoriasis via an attenuation of the local deregulation of several cytokines/chemokines, including IL-6 and vascular endothelial growth factor-A isoforms, and of a concurrent, abnormal cell differentiation program entailing the expression, amongst other proteins, of cytokeratin-16 [27] studies

conducted by the same research group investigated the effects of CTSW on cultures of epidermal keratinocytes isolated from the lesion areas of nine psoriatic patients, concluding that CTSW exposure significantly downregulated the intracellular levels of TNF-$\alpha$, and also reduced the intracellular levels and secretion rates of IL-8 [29].

Other studies focused on the effects of CTSW in skin regeneration. In an experimental study, Faga et al. [30] investigated the effect of the CTSW on the wound healing process in comparison with other treatments: (a) monolayer petrolatum gauze, two layers of sterile nonwoven fabric gauze and a transparent film dressing; (b) the same plus gauzes soaked in sterile solution; (c) the same plus gauzes soaked in CTSW. The results showed that CTSW improved skin regeneration, not only by increasing keratinocyte proliferation and migration but also favorably modulating the regenerated collagen and elastic fibers in the dermis.

Based on the previous studies, the authors considered that the skin regeneration results, and other biological properties, may be not entirely explained by chemical composition. Thus, Nicolleti et al. [64] supposed that the nonpathogenic bacterial within the CTSW could be correlated with the demonstrated biological properties of this water and the results appointed in this involvement of the bacteria. Later, in an in vitro trial, these authors evaluated the effects of the same spring water on human skin fibroblasts and found that the cell viability of the cultures was higher when cultivated with 10% and 20% of CTSW in comparison with the controls. They suggested that these effects could be due to the presence of nonpathogenic bacterial flora, and the whole spring water microbiota might be responsible for producing molecular mediators with a role in the wound healing process [65].

The same research group also conducted a study to evaluate the efficacy of spring water treatment in an ex vivo human skin model for assessment of wound healing. In this study, filtered CTSW was used, supposing that metabolites from the microbiota remained in the filtered water. The study identified favorable biological events in the human skin samples treated with filtered CTSW. The most notable effects were evident in the dermis, showing a markable anti-inflammatory effect by reducing overall dermal cell infiltration when compared with the controls. The authors postulated that the reduction in cellular infiltrate in the dermis was concomitant with fibroblast recruitment, suggesting a favorable modulation of the local cell proliferative phase. The authors concluded that these effects are potentially associated with the functions of the active metabolites produced by the spring water's native microbiota, and preparations with CTSW may have clinical efficacy in promoting both tissue regeneration and wound healing [26].

Afterwards, to confirm this hypothesis, this group investigated the action of selected bacterial lysates that were isolated from CTSW on an in vitro culture of human skin fibroblasts. Human fibroblast cultures were used adding four bacterial lysates and cell proliferation was evaluated by spectrophotometric absorbance analysis after the XTT-microculture tetrazolium assay. The results showed a remarkable stimulation of cell proliferation only after addition of Firmicute-derived bacterial lysates to the culture medium. The authors concluded that the study showed a favorable action of the native microbiota in CTSW on human fibroblast proliferation, and a combination of biological properties of several bacterial species within this spring water might be responsible for its regenerative effects on human skin [31].

### 4.4. Cró Thermal Spring Water

Cró thermal water is a medium mineral thermal water, from a thermal spa in the Beira Interior region (Portugal), whose main components are sodium, silica, calcium, potassium and some trace elements (Table 5) [32].

**Table 5.** Cró thermal water composition (adapted from [32]).

| Anions/Cations/Trace Elements and Other Compounds | |
|---|---|
| Bicarbonate (mg/L) | 157 |
| Sulphate (mg/L) | 14.1 |
| Chloride (mg/L) | 33 |
| Nitrate (mg/L) | <0.20 |
| Fluoride (mg/L) | 15.7 |
| Silica $SiO_2$ (mg/L) | 47.8 |
| Sodium (mg/L) | 103 |
| Calcium (mg/L) | 3.5 |
| Magnesium (mg/L) | 0.21 |
| Potassium (mg/L) | 2.7 |
| Lithium (mg/L) | 0.35 |

Nunes et al. [32] developed a dermocosmetic formulation (hydrophilic gel), comprising more than 90% of thermal water, compared it with a control (a gel formulation with identical composition but using purified water) and performed characterization and stability studies and efficacy evaluation using noninvasive biometric techniques. The Cró thermal water was biocompatible, since normal human dermal fibroblasts (NHDF) adhered and proliferated in the presence of this water. The developed dermocosmetic formulation during the period of storage showed adequate organoleptic properties. The gel prepared with thermal water showed lower firmness, adhesiveness and spreadability values than the control gel, showing acceptable characteristics for skin topical application. It is to be assumed that the thermal water-based gel is expected to be easier to spread onto the skin and to allow a sustained release of active compounds. Twenty healthy human volunteers participated in the study to evaluate the effect of the formulation on different parameters, such as skin pH and degree of hydration transepidermal water loss (TEWL). Results showed that after gel application, the preservation of the physiologic acid nature of the skin is adequate. The results showed that the hydration degree was higher in application of thermal water-based gel formulation, confirming the improvement of the incorporation of Cró mineral water in the formulation. After 30 min, a statistically significant decrease in the TEWL was registered, which suggests that this formulation could have some occlusive effect on skin barrier function. Furthermore, this formulation showed higher lesser roughness in the surface evaluation, as well as lower scaliness and higher smoothness. This work shows the beneficial effects of this thermal water in dermocosmetic formulations and is expected to be a potential tool for the treatment of dermatological diseases (Table 1).

*4.5. Dead Sea Thermal Water*

Dead Sea, which composition rich in magnesium, calcium, sodium, potassium, zinc and strontium are known for their therapeutic efficacy in treating a variety of skin conditions, as psoriasis, AD, vitiligo, acne and eczema, as well as for their cosmetic benefits [33,35,66] (Table 6).

An in vitro study conducted by Meital Portugal-Cohen et al. [33] tested Dead Sea mineral-rich water (DSW) and anionic polysaccharide (PolluStop®) separately and combined in different proportions to protect the skin against damage caused by urban pollution, being evaluated using a normal human-derived epidermal keratinocyte culture and assessed by the inflammatory biomarkers IL-1α and prostaglandin E2, in addition to epidermal viability. After exposure to mixture of pollutants, containing heavy metals and atmospheric particulate matter that caused inflammation of the epidermis, the treatments with DSW, PolluStop® and their combinations, in different ratios, presented a significant inhibition in IL-1α release. DSW 0.25% alone showed the highest IL-1α inhibition. None of the test materials and their combination fully inhibited the sharp decrease in epidermal cell viability. Ozone exposure of the epidermis resulted in a significant increase in the release of both IL-1α and prostaglandin E2, along with a mild decrease in tissue viability. Only when DSW and PolluStop® were mixed was significant inhibition of these inflammatory

markers observed, being that the combination of DSW 0.8% + PolluStop® 5% is the only one capable of favor against mixture of pollutants and ozone pollutants, as it is capable of inhibiting all inflammatory markers tested.

**Table 6.** Dead Sea thermal water composition (adapted from [67,68]).

| Anions/Cations/Trace Elements and Other Compounds | |
|---|---|
| Chloride (mg/L) | 224,200 |
| Bicarbonate (mg/L) | 200 |
| Sulphate (mg/L) | 280 |
| Bromide (mg/L) | 4500 |
| Calcium (mg/L) | 17,600 |
| Magnesium (mg/L) | 42,120 |
| Potassium (mg/L) | 7600 |
| Sodium (mg/L) | 41,600 |
| Strontium (mg/L) | 150 |
| Manganese (µg/L) | 9800 |
| Zinc (µg/L) | 16 |
| Nickel (µg/L) | 41.08 |
| Copper (µg/L) | 2.18 |
| Cobalt (µg/L) | 0.69 |

In another in vitro study by Meital Portugal-Cohen et al. [35], the ability of a patented leave-on skin emulsion enriched with DS mineral mud and with DSWand other actives (Dermud$^{TM}$) to protect human skin against UVB-induced biological effects was evaluated. For this study, human skin cultures were used as a model, more specifically, skin fragments obtained from healthy women aged between 20 and 60 years. Dermud$^{TM}$ cream was applied on to the air exposed epidermis. UVB exposure caused a 20% decrease in mitochondrial activity 48 h after irradiation, but previous application of Dermud$^{TM}$ reversed this trend, with a 20% increase in the same conditions, and it also caused a drastic 10-fold increase in caspase 3 activity, but a previous treatment with Dermud$^{TM}$ reduced this increase by 80%. Topical treatment by Dermud$^{TM}$-enhanced ferric-reducing antioxidant power (FRAP) by 50% in nonirradiated controls, and the Dermud$^{TM}$-enhanced antioxidant power was not affected by UV irradiation. When Dermud$^{TM}$ was applied the UVB-derived, low molecular weight, antioxidant-consuming ROS, released by skin samples, were totally neutralized. The Dermud$^{TM}$ also helped in acid uric depletion, after UVB irradiation, with the acid uric partially restored by the cream pretreatment. Dermud$^{TM}$ application prior to irradiation drastically inhibited the secretion of inflammatory cytokines in skin samples: TNF-$\alpha$ and IL-1$\alpha$, IL-6 and IL-8. In this study, it was demonstrated that Dermud$^{TM}$ reduced skin photodamage and photoaging and more generally reduced oxidative stress and inflammation in skin pathologies through antioxidant, antiapoptotic and anti-inflammatory properties.

A 12-week, double-blind controlled study was conducted by Meital Portugal-Cohen et al. [34] on 86 children with mild-to-moderate AD to evaluate the efficacy of an emollient cream enriched with thermal water from the Dead Sea. The patients were randomized to receive twice-daily topical treatment with a cream enriched with Dead Sea minerals, compared to two types of control: (1) a cream with Dead Sea minerals with concentrations of DSW lower than the previous one and a Dead Sea mineral-free emollient cream. The study showed that both creams with Dead Sea minerals improved objective severity assessment of atopic dermatitis (OSAAD) scores. Only cream enriched with Dead Sea minerals improved TEWL and stratum corneum hydration. The cream enriched with Dead Sea minerals was the most effective regarding TEWL, stratum corneum hydration and OSAAD compared to the creams tested. As among the treatment groups, cream enriched with Dead Sea minerals efficacy was the highest in most severity scores, we can conclude that Dead Sea water plays a key role in improving the skin barrier function and can be used as an effective adjuvant treatment in AD (Table 1).

### 4.6. La Roche-Posay Thermal Spring Water

La Roche-Posay Thermal Spring Water (LRPTSW) (France) is a medium mineral water that contains a particularly high content of selenium [11,42] (Table 7).

**Table 7.** La Roche-Posay thermal spring water composition (adapted from [42,69]).

| Anions/Cations/Trace Elements and Other Compounds | |
| --- | --- |
| Bicarbonate (mg/L) | 387 |
| Sulphate (mg/L) | 56.1 |
| Chloride (mg/L) | 22.6 |
| Nitrate (mg/L) | 1.6 |
| Fluoride (mg/L) | 0.2 |
| Bromide (mg/L) | 0.3 |
| Phosphate (mg/L) | <0.1 |
| Silica $SiO_2$ (mg/L) | 31.6 |
| Calcium (mg/L) | 149.0 |
| Magnesium (mg/L) | 4.4 |
| Potassium (mg/L) | 1.9 |
| Sodium (mg/L) | 1.3 |
| Lithium (mg/L) | <0.1 |
| Iron (mg/L) | <0.005 |
| Manganese (mg/L) | <0.003 |
| Strontium (mg/L) | 0.3 |
| Selenium (µg/L) | 53 |
| Copper (µg/L) | <5 |
| Zinc (µg/L) | <5 |

To evaluate the antiradical properties of LRPTSW, in vitro studies were performed in which human skin fibroblasts were cultured and exposed to oxidative stress in the presence or absence of LRPTSW. After exposure to UVB and hydrogen peroxide, selenium-glutathione peroxidase and superoxide dismutase activity was higher in fibroblasts cultured in medium containing LRPTSW than in other media. After exposure to UVA, in the medium with LRPTSW, the selenium–glutathione peroxidase activity and cell viability were significantly increased [39]. Furthermore, after exposure to increasing UVB doses (50 to 200 mJ/cm$^2$), human keratinocytes cultured in medium containing LRPTSW showed better resistance with better cell survival and a reduction in IL-1 release in comparison with cells cultured in medium containing demineralized water [40].

In a study carried out on human keratinocytes, after exposure to increasing UVB doses, showed better resistance with better cell survival and a reduction in IL-1$\alpha$ cytokine release in cultured in medium containing LRPTSW in comparison with cells cultured in medium containing demineralized water. An in vitro study evaluated the modulatory effects of media containing selenium or strontium salts, as well as LRPTSW. These effects were evaluated using a reconstructed model of biopsies of healthy skin or skin with atopic dermatitis, for the production of IL-1$\alpha$, IL-6 and TNF-$\alpha$. In the three media, after 10 days of culture, the production of these three cytokines was shown to be lower in inflamed skin than in healthy skin. With strontium salts its selective inhibitory effect on IL-1$\alpha$ production was less evident, but the inhibitory effect for TNF-$\alpha$ was greater. At intracellular and extracellular levels, selenium and strontium salts showed a modulating effect in decreasing IL-6 production [37].

The effect of selenium from LRPTSW has also been studied in cultured human skin fibroblasts. Cells were cultured in a medium containing 2% fetal calf serum and supplemented with selenium or in LRPTSW and compared with control medium containing demineralized water. A protective effect on lethality of dividing fibroblasts induced by UVA radiation was found, as well as an increase in glutathione peroxidase activity [36]. Later studies confirmed these effects; in a study conducted by Zöller et al. [11] comparing different waters, LRPTSW was able to reduce IL-6 levels and the formation of ROS after UVB irradiation. The authors suggested that the observed effects could be related

to the high selenium content (53 μg/L), which is a co-factor for glutathione peroxidase, a key-enzyme in the elimination of ROS.

Furthermore, in an in vitro study with UVB-irradiated keratinocytes and fibroblasts, LRPTSW was compared with demineralized water; results showed that LRPTSW was able to reduce IL-1α release in irradiated keratinocytes. Moreover, an increase in the selenium-dependent glutathione peroxidase activity and a decrease in the production of lipoperoxides in irradiated fibroblast was observed. The authors suggested that selenium-rich thermal water could play an important role in the prevention of the UV damage [38].

A comparative study carried out by Cadi et al. [43] evaluated anticancer properties by assessing the reduction in the number of UVB-induced skin tumors. In this study, a significant reduction in the number of UVB-induced skin tumors, along with an increased lag time until first tumor appearance; malondialdehyde formation was noticed and significant increase in selenium–glutathione peroxidase activity was reached in a group treated with a cream containing LRPTSW before chronic exposure to UVB. In addition, glutathione peroxidase activity remained stable in control rodents and irradiated rodents not treated, whereas in the group treated with LRPTSW cream, glutathione peroxidase activity increased significantly during treatment.

A comparative study carried out with a sample of 10 volunteers consisted of applying sodium lauryl sulphate (concentration equal to 0.75%) under occlusive conditions for 24 h on the forearm of the volunteers and evaluating the cutaneous blood flow, before and after treatment, with a gel containing LRPTSW and another gel containing demineralized water. Blood flow was reduced by 46% and 15%, respectively, thus supporting the anti-inflammatory effect of water [42].

A randomized double-blind study was performed on 10 men and women to assess the ability of a cream containing LRPTSW to protect against UVB-induced erythema. Two creams, one formulated with LRPTSW and one with deionized water, were applied and the treated areas were exposed to increasing doses of UVB. After 24 h, the number of burned cells per square centimeter of epidermis was significantly reduced in the areas pretreated with the cream containing LRPTSW, compared to the same cream containing demineralized water [70].

In addition, one observational study should be mentioned, related to scars. Two cases of use of spray with LRPTSW in the treatment of scars after pediatric plastic surgery obtained positive results, where there was a softening of the inflammatory appearance of the same, reduction of pruritus, as well as facilitation of the elimination of crusts and prevention the formation of new crusts [42].

### 4.7. Monfortinho Thermal Water

Monfortinho thermal spring water (MTSW) has low mineral content; sodium and silica represent more than 50% of total mineralization [44,45] (Table 8).

Almeida et al. [44] carried out a study on MTSW to explore its potential use in dermocosmetic formulations and its effectiveness in cutaneous disorders. To accomplish this, three creams (O/W emulsions) were prepared, two prepared with thermal water at two concentrations and one with ultrapure laboratory water, and their organoleptic characteristics, viscosity and rheological profile were evaluated, with the results showing that the formulations are compatible with skin.

Next, a double-blind study on 30 patients with psoriasis, AD and eczema with different stages of the disease was designed to assess the barrier function of the skin after being applied the formulations twice a day for 28 consecutive days. In biometric analysis of the skin, with regard to epidermal capacitance and the TEWL, the data obtained did not reveal any significant differences between the three creams tested. The data revealed an incidence of positive results or qualitative improvement of erythema. The highest positive percentage was the one formulated with laboratory ultrapure water. Regarding the results of flaking symptoms, these were more positive in the cream with MTSW. All creams benefitted from the prevention of chafing and dryness of skins. With the cream

formulated with MTSW, an effect more positive for itching and flaking skin disorders was observed. Although statistically significant differences when comparing the tested formulations were not observed, this study showed that MTSW has a positive effect on the skin and dermatological diseases [44].

**Table 8.** Monfortinho thermal spring water composition (adapted from [44]).

| Anions/Cations/Trace Elements and Other Compounds | |
|---|---|
| Sulphate (mg/L) | <5 |
| Sulfide (mg/L S) | <0.17 * |
| Chloride (mg/L) | 3.7 |
| Dissolved carbon dioxide (mg/L $CO_2$) | 27 |
| Nitrate (mg/L) | <5.0 * |
| Fluoride (mg/L) | 0.05 |
| Phosphate (mg/L) | <0.05 * |
| Silica $SiO_2$ (mg/L) | 16 |
| Calcium (mg/L) | 1.6 |
| Magnesium (mg/L) | 2.7 |
| Potassium (mg/L) | 0.90 |
| Sodium (mg/L) | 3.3 |
| Boron (mg/L) | <0.17 * |
| Manganese (µg/L) | <15 * |
| Zinc (µg/L) | <0.20 * |
| Cobalt (µg/L) | <4 * |
| Copper (µg/L) | <10 * |
| Chromium (µg/L) | <10 * |
| Selenium (µg/L) | <1 * |

* < LOQ, limit of quantification.

A more recent study by Oliveira et al. [45] investigated the in vitro effect of MTSW on the homeostasis of skin cells, cultivating two representative cell lines of the epidermis (keratinocytes) and dermis (fibroblasts) and, in addition to a mouse macrophage cell line, in a culture medium prepared with MTSW.

After exposure to MTSW, compared to the control, the compromise in metabolic activity decreased in keratinocytes and in fibroblasts by 60% and 45%, respectively, as well as the occurrence of a decrease in cell proliferation in both. In fibroblasts, MTSW also induced a slight decrease in cell migration and significantly increased the number of etoposide-induced senescent cells. Regarding macrophages, MTSW produced a significant decrease in cell metabolism.

The reduction in both cell metabolism and proliferation of keratinocytes and macrophages highlight the benefits of MTSW in hyperkeratotic conditions, such as psoriasis and atopic dermatitis (Table 1) [45].

### 4.8. Saint-Gervais Mont Blanc Thermal Spring Water

Saint-Gervais Mont Blanc thermal spring water (SGMBTSW) is naturally strongly mineralized with high rates of calcium and other trace elements such as manganese, boron and zinc (Table 9). It is well-known for its healing dermatological properties, and is used in the treatment of psoriasis, eczema and burn scars [71]. The benefits of treatment developed in the Saint-Gervais Mont Blanc thermal center on scar sequelae following burns was presented in the review "La Presse Thermale et Climatique" in 1964, 1974 and 1985 [72–74], but these references only can be found in the Semantic Scholar database.

**Table 9.** Saint-Gervais Mont Blanc thermal spring water composition (adapted from [71]).

| Anions/Cations/Trace Elements and Other Compounds | |
|---|---|
| Sulphate (mg/L) | 1812 |
| Chloride (mg/L) | 530 |
| Bicarbonate (mg/L) | 247 |
| Sodium (mg/L) | 944 |
| Calcium (mg/L) | 234 |
| Magnesium (mg/L) | 26.8 |
| Potassium (mg/L) | 29 |
| Barium (mg/L) | 17 |
| Manganese (mg/L) | 0.327 |
| Boron (mg/L) | 5.03 |
| Strontium (mg/L) | 8.9 |
| Zinc (μg/L) | 57 |
| Iron (μg/L) | <30 |
| Aluminium (μg/L) | <15 |
| Antimony (μg/L) | <5 |
| Copper (μg/L) | <2 |

In a review by Gravelier et al. [75], to provide an overview of the spa therapy used in the treatment of burn scars, the authors described two clinical studies [76,77] that concluded that spa therapy provides beneficial effects in burn scar recovery; despite the effective public health practice project quality assessment tool for quantitative studies being applied, the quality rating was weak for both studies [75].

To understand the role of manganese and boron in wound healing observed in SGMBTSW, Chebassier et al. [47] investigated the modulation of keratinocyte migration and proliferation using boron and manganese salts in an in vitro study. They incubated keratinocytes obtained from children's foreskins for 24 h with boron salts at concentrations between 0.5 and 10 mg/mL or manganese salts at concentrations between 0.1 and 1.5 mg/mL and demonstrated that these salts accelerated wound closure compared with the control medium (+20%). They found that this acceleration was not related to an increase in keratinocyte proliferation; thus, the authors suggested that boron and manganese act on wound healing mainly by increasing the migration of keratinocytes.

In order to evaluate the ability of SGMBTSW to improve both keratinocyte differentiation and barrier function, human epidermal keratinocytes, obtained from plastic surgery procedures, were treated with increasing concentrations of SGMBTSW thermal water, with manganese gluconate or with a combination of both. The results showed that SGMBTSW spring water stimulates transcriptomic expression of key markers involved in keratinocyte differentiation and barrier function, while manganese gluconate has no effect. Combination of both dramatically enhances keratinocyte differentiation, in a synergistic way, at both the transcriptomic and protein level. None of the treatments modulated ATP2C1 expression. The authors concluded that these results highlight the interest in enriching SGMBTSW spring water with manganese to boost keratinocyte differentiation and barrier function [71].

*4.9. Salies-de-Béarn Thermal Spring Water*

Salies-de-Béarn thermal spring water (SBTSW) is a very high saline natural water, rich in sodium and magnesium [48] (Table 10). Despite being a very well-known thermal water for the treatment of several diseases in the areas of rheumatology, gynecology and pediatrics, there are no scientific studies about its effects. In addition, this thermal water is sold in the form of concentrated salt water ("eaux meres") to prepare cosmetics [78], but there is a lack of studies about its effects.

**Table 10.** Salies-de-Béarn thermal spring water composition (adapted from [48,79]).

| Anions/Cations/Trace Elements and Other Compounds | |
|---|---|
| Chloride (mg/L) | 152,140 |
| Sulphate (mg/L) | 980 |
| Carbonate (mg/L) | 225 |
| Magnesium (mg/L) | 980 |
| Potassium (mg/L) | 1370 |
| Calcium (mg/L) | 1450 |
| Fluoride (mg/L) | 1 |
| Bromide (mg/L) | 160 |
| Silicon (mg/L) | 30 |
| Lithium (mg/L) | 9 |
| Iron (mg/L) | 32 |
| Manganese (mg/L) | 8 |
| Boron (mg/L) | 1.5 |
| Chromium (mg/L) | 2 |

In the search process whilst carrying out this literature review, we only found one article related to the therapeutic effect of SBTSW. Léauté-Labrèze et al. [48], in a clinical study, investigated the efficacy of this thermal water on psoriasis, resulting in the conclusion that saline spa water alone had a minor therapeutic effect in psoriasis, and the beneficial effect of bathing to enhance phototherapy was not demonstrated.

*4.10. São Pedro do Sul Thermal Water*

São Pedro do Sul thermal spring water (SPSTSW) is a medium mineral water. Analysis of its chemical composition revealed the presence of bicarbonate, sodium and silica with especially high sulfur content [46] (Table 11).

**Table 11.** São Pedro do Sul thermal spring water composition (adapted from [46]).

| Anions/Cations/Trace Elements and Other Compounds | |
|---|---|
| Bicarbonate (mg/L) | 126.9 |
| Sulphate (mg/L) | 9.7 |
| Chloride (mg/L) | 27.7 |
| Fluoride (mg/L) | 18.2 |
| Carbonate (mg/L) | 4.8 |
| Silicate (mg/L) | 12.4 |
| Nitrate (mg/L) | <0.12 |
| Total sulfur (in $I_2$ 0.01 N) (mg/L) | 18.4 |
| Silica $SiO_2$ (mg/L) | 65.5 |
| Calcium (mg/L) | 3 |
| Magnesium (mg/L) | <0.03 |
| Potassium (mg/L) | 3.3 |
| Sodium (mg/L) | 93 |
| Selenium (µg/L) | <0.0012 |

In a study performed by Ferreira et al. [46], the effect of the SPSTSW on skin irritation induced by sodium lauryl sulphate was investigated. A set of 17 healthy Caucasian volunteers aged 21–42 years were enrolled in this study. On each of the volunteers' forearms, two skin sites were marked and irritated with sodium lauryl sulphate at 2% (*w/v*). SPSTSW or purified water was applied and tested in equal damaged forearm skin sites. The SPSTSW showed a statistically significant anti-irritant effect when, after the application of both waters, it reduced the degree of skin barrier breakdown in 82.4% of the volunteers compared to laboratory-purified water. Thus, it is concluded that SPSTSW has an anti-irritant effect, assuming that anionic ions are mainly responsible, and can be used in cosmetic products, increasing the tolerability of these formulations.

*4.11. Uriage Thernal Spring Water*

Uriage thermal spring water (UTSW) (France) is an isotonic water particularly rich in sodium, calcium, magnesium and silica and with trace elements such as zinc and copper (Table 1) [49–51] (Table 12).

**Table 12.** Uriage thermal spring water composition (adapted from [49–51]).

| Anions/Cations/Trace Elements and Other Compounds | |
|---|---|
| Sulphate (mg/L) | 2860 |
| Chlorides (mg/L) | 3500 |
| Sodium (mg/L) | 2360 |
| Bicarbonate (mg/L) | 390 |
| Calcium (mg/L) | 600 |
| Magnesium (mg/L) | 125 |
| Potassium (mg/L) | 45.5 |
| Silicon (mg/L) | 42 |
| Zinc (mg/L) | 0.160 |
| Manganese (mg/L) | 0.154 |
| Copper (mg/L) | 0.075 |
| Iron (mg/L) | 0.015 |

A study developed by Verdy et al. [49] investigated the effect of UTSW on skin protection by evaluating taurine transporter (TauT) and sodium-dependent vitamin C transporter 1 (SVCT1) expressions in normal human keratinocytes at a basal level and under stressful conditions (irradiation UVB). For the study on TauT expression, keratinocytes were incubated with UTSW at increasing concentrations after irradiating the cells with UVB, and the effect of UVB was slightly reduced after 24 h and maintained a high level of TauT expression after 72 h. To evaluate SVCT1 expression, firstly, the SVCT1 in keratinocytes issued from two "young" and two "aged" donors was quantified, and the results showed high expression levels of SCVT1 in "young" donors, enabling the explanation of the inverse correlation in the skin between vitamin C and increasing age. In the next step, the effect of UTSW on SVCT1 expression was studied; the cells were incubated in the absence (control) and in the presence of increasing concentrations of UTSW. In "young" keratinocytes, a decrease in SVCT1 expression and in cells issued from "aged" donors occurred; the decreased SVCT1 expression was partially reversible. The UTSW effect on the UVB irradiated on normal human keratinocyte models was also evaluated. It was observed that UTSW at 1% (*v/v*) interferes with the mechanisms associated in the UVB-repressed expression of SVCT-1 in "young" cells, providing a protective effect against solar irradiation, and as the effect of the thermal water on SVCT-1 expression in "aged" cells did not persist, UTSW is not strong enough to counteract the effects of aging and UVB irradiation. Regarding TauT, UTSW did not significantly change the basal level of TauT expression in normal human keratinocytes, except for the higher UTSW concentration tested at 72 h. When the cells were irradiated by UVB, TauT expression increased after 24 h and returned to basal level after 72 h. In these conditions, UTSW slightly reduced the effect of UVB after 24 h of incubation, but above all, permitted the maintenance of a high level of TauT expression after 72 h of incubation. In conclusion, these results suggested that UTSW could act to efficiently protect the skin from dehydration through its effect on TauT and SVCT1 expression, and to furthermore permit a more efficient taurine and vitamin C supply to the epidermis in order to protect it from other aggressions such as oxidant stress. In addition, the UTSW could also be useful for the skin in fighting against situations such as dehydration, UVB irradiation and aging (Table 1).

A study conducted by Joly et al. [51] evaluated the advantage of one antidry skin product cosmetic, formulated with thermal water instead of deionized water through the evaluation of the epidermal expression of aquaporine-3 (AQP-3), filaggrin, claudin-4 and claudin-6 in human skin explants of stratum corneum. When explants were incubated with the cosmetic product formulated with deionized water after the delipidation procedure, a

slight effect on the stratum corneum reparation was obtained, but the filaggrin labelling remained poor and discontinuous and with regard to AQP-3 expression, it only had a weak effect. When explants were treated using the cosmetic product formulated with UTSW, a very good effect on stratum corneum reparation was observed, and filaggrin labelling and AQP-3 labelling were equivalent to that observed in the control explants. This cosmetic is also able to enhance the expression of claudin-4 and claudin-6 in the delipidated explants. This study showed that the use of UTSW in cosmetic formulations contributes to increasing their efficacy in dry skin conditions (Table 1).

More recently, Joly et al. [50] carried out a study to evaluate the protective effects of the UTSW on the skin barrier function at three levels: antioxidant properties through thiobarbituric acid-reactive substances and superoxide dismutase activity assays, using an in vitro model of NHDF cells treated with a hypoxanthine/xanthine oxidase (HX/XO) mixture; the protective role on nuclear DNA damage, using human keratinocytes irradiated or not by UVB; and catalase activity determination and cutaneous claudin-6 expression, using an ex vivo model of human skin explants irradiated (or not) with UVA and UVB. It was observed that UTSW significantly and dose-dependently increase the survival rate of NHDF cells submitted to an HX/XO stress. UTSW was able, at the same time, to reduce the lipid peroxidation induced by HX/XO stress. Additionally, UTSW manifests considerable SOD-like activity. The skin explants' intracellular catalase activity was dramatically and effectively reduced by the cosmetic formulation containing UTSW, allowing the skin to partially regain its catalase activity in 75% of the cases. Under the same experimental conditions, UVB caused a considerable increase in DNA degradation in the study of DNA damage, but UTSW lessened this damage. Skin claudin-6 expression was significantly reduced when skin explants were exposed to UVA + UVB radiation, but it was restored when a cosmetic product containing UTSW was previously applied to the skin's surface. This strategy highlights UTSW's antioxidant action and its part in DNA protection. This water could also help in fighting against UV-induced decrease in the putative tumor suppressor claudin-6 (Table 1).

Gannesen et al. [52] studied the effect of commonly used cosmetics, UTSW and a rhamnose-rich polysaccharide (PS291®) on acneic strains (RT4 and RT5) of *Cutibacterium acnes* and a cutaneous strain of *S. aureus*. UTSW affected the growth kinetic of acneic *C. acnes* by essentially increasing its generation time and reducing its biomass, whereas only the *S. aureus* final biomass was reduced, while PS291® had more limited effects in the biofilm formation. The association of UTSW and PS291 only induced a partial reduction in *S. aureus* initial adhesion of the same range observed using PS291®, as if, in regard to this process, the two cosmetic compounds had antagonistic effects. Both compounds showed a marked antibiofilm activity on *C. acnes* and *S. aureus*. For *S. aureus* that essentially appeared due to inhibition of initial adhesion, *C. acnes* and *S. aureus* showed clear hydrophobic surface properties. UTSW and PS291® had a restricted effect on *C. acnes* but increased the hydrophobic character of *S. aureus*. The cosmetics did not modify the metabolic activity of bacteria. In general, this study underlines the effect of cosmetics on these cutaneous bacteria.

Taking into account that UTSW interferes with *C. acnes* biofilm formation, the effect of catecholamines on acneic (RT4) and nonacneic (RT6) strains of *C. acnes* in medium with UTSW compared with sterile physiological water was evaluated. On an RT4 strain grown in a sterile physiological medium after exposure to epinephrine, the formation of biofilm was observed, while it was observed that lower basal production of biofilm with UTSW and exposure to epinephrine resulted in biofilm formation that remained significantly lower than that of bacteria grown in medium containing sterile physiological water. The response of RT4 *C. acnes* to norepinephrine was remarkably similar; however, in the presence of sterile physiological water medium, stimulated biofilm development at a greater rate, and the UTSW was unable to entirely inhibit this effect. On the other hand, UTSW decreased RT6 biofilm formation, but exposure of the bacterium to epinephrine in the presence of UTSW induced a limited but significant increase in biofilm. Hence, the effect of UTSW on

the response of *C. acnes* to catecholamines depended on the surface on which the biofilm was grown and suggested that *C. acnes* may play a role as a relay between stress mediators (catecholamines) and acne [53].

*4.12. Vichy Thermal Spring Water*

Vichy thermal spring water (VTSW) is a very high minerality water that contains particularly high calcium content (Table 13) [54].

**Table 13.** Vichy thermal spring water composition (adapted from [54]).

| Anions/Cations/Trace Elements and Other Compounds | |
| --- | --- |
| Hydrogenocarbonates (mg/L) | 4818.633 |
| Orthophosphate (mg/L) | 0.210 |
| Sulphate (mg/L) | 182.39 |
| Boron (mg/L) | 0.970 |
| Calcium (mg/L) | 165.61 |
| Fluoride (mg/L) | 7.67 |
| Lithium (mg/L) | 5.17 |
| Magnesium (mg/L) | 12.08 |
| Potassium (mg/L) | 103.56 |
| Silicon (mg/L) | 11.78 |
| Sodium (mg/L) | 1862.88 |
| Strontium (mg/L) | 1.63 |
| Iron (mg/L) | 0.810 |
| Manganese (mg/L) | 0.208 |

Tacheau et al. [54] examined the effects of the VTSW on human keratinocytes grown in vitro them in with or without VTSW for 18 h. In keratinocytes incubated with VTSW, the expression of collagen VII, laminin 5, α6β4 integrin and nidogen 1 was stimulated, increasing basement structure and adhesive capacity. It also increased the expression of two markers of cell proliferation (Ki67 and proliferating cell nuclear antigen), and heparin-binding epidermal growth factor-like growth factor (HBEGF), involved in wound re-epithelialization. Modulators of proliferation–differentiation were induced (CD44, stratifin and AQP3), pointing out that in the presence of VTSW, keratinocyte proliferation is stimulated, but under the control of key actors. The stimulated expression of CK10, TGM-1 and FLG demonstrates that the activation of proliferation and differentiation is indeed balanced in the presence of VTSW. The desmosomes involved in keratinocyte adhesion and epidermal integrity were also stimulated in the presence of VTSW. The expression of cathepsin V, L, D, KLK5 and KLK7 was also stimulated, thus favoring cross-linked envelope formation and stratum corneum desquamation process. The increased expression in cystatin M/E, an inhibitor of cathepsins, suggests that cross-linked envelope formation and desquamation may be triggered by VTSW, and that it would be in a controlled and balanced way. In the presence of VTSM antioxidant mechanisms were potentially mobilized in a coordinated manner through the stimulation of 14 genes, 5 of them corresponding to antioxidant enzymes regulated by the Keap1–Nrf2 pathway. The expression of a set of 12 genes involved in DNA repair and response to stress-induced DNA damage was increased in the presence of VTSW. Globally, the presence of VTSW increased the expression of a number of skin homoeostasis-related genes. Thus, it is possible that VTSW could be an active ingredient in cosmetic formulations aimed at addressing some of the harmful effects of skin-ageing exposomes (Table 1).

Gueniche et al. [55] evaluated the effects on skin functions of an extract of *Vitreoscilla filiformis* grown in a medium containing VTSW with probiotic properties and evaluated it in combination with VTSW. For the evaluation of the effects in skin barrier function, normal human epidermal keratinocyte cells were used, and the combination of 10% VTSW and extract compared with the controls significantly increased transglutaminase, filaggrin, involucrin, claudin-1, and zonula occludens-1. Regarding antimicrobial peptide defenses, a

combination of VTSW and extract increased the expression of β-defensin-4A and S100A7. In terms of skin immune defense functions, in lipopolysaccharide-stimulated peripheral blood mononuclear cells, the combination of VTSW and extract was shown to downregulate IL-8, TNF-α and IL-12/IL-23p40 and increased the IL10 and IL-10/IL-12 ratio. Additionally, the combination of VTSW plus 5% extract protected Langerhans cells in skin explants exposed to UV radiation. Thus, the combination of extract plus VTSW has properties that reinforce the skin barrier by stimulating skin differentiation and tight junctions, biochemical defenses through the stimulation of antimicrobial peptides and cellular immune defenses by increasing the IL-10/IL-12 ratio and by protecting Langerhans cells challenged by UV radiation.

In another study by the same author, in vivo and ex vivo assays with a dermocosmetic formulation containing 80% VTSW, 5% extract, 4% vitamin B3, 0.4% hyaluronic acid and 0.2% vitamin E, known as Minéral 89 Probiotic Fractions (M89PF) were presented to evaluate clinical efficacy in preventing and repairing stressed skin. Regarding skin barrier effects, the dermocosmetic formulation significantly enhanced skin renewal compared to untreated skin after exposure to skin previously exposed to sudden thermal changes after skin irritation by tape stripping and in sleep-deprived women. In terms of skin antioxidant defense activity, the microbiome recovery after acute stress from a harsh cleanser was significantly improved in formulation-treated skin compared to bare skin. The dermocosmetic formulation also showed depigmenting properties on dark spots in women exposed to a stressful lifestyle and various external (pollution, tobacco smoking, solar radiation) and internal (poor sleep, stressful work, unbalanced diet and alcohol consumption) exposome factors. This study enabled the understanding of the clinical benefits of the dermocosmetic formulation. Even so, clinical trials are crucial in assessing the efficacy of formulation as adjuvant care to prevent and repair skin barrier disruption and reinforce defenses in skin exposed to acute stress [56].

## 5. Incorporation of Thermal Spring Waters in Cosmetic Formulations

The biological effects of thermal waters are associated with its chemical composition and some physical properties, especially the concentration of the predominant compounds, as well as the presence of some trace elements [42,44,50]. The beneficial effects of the treatment of some skin diseases are fundamental to the use of thermal waters as an active ingredient in dermocosmetic formulations [5]. Despite this, there are few studies that compare the activity of thermal spring water included in cosmetics as an active ingredient. Some compare two types of thermal spring water, mainly if their mineralization is very different (low mineralization versus high mineralization), but never more than two.

Thermal water, in natura or manipulated in products, has been described in dermatology as an adjunct to the hydration of the skin, in the treatment of skin aging, acne, rosacea and other inflammatory dermatoses, and after cosmetic procedures, such as chemical peels, laser, etc., through improvement to the skin's properties in terms of hydration, flexibility and elasticity, and exerting different effects, such as anti-inflammatory, calming, desensitizing, healing and antioxidant (Table 1) [3–5,32,50]. Currently, cosmetic products developed based on thermal water include sprays that convey fresh water, as well as products developed based on thermal water, as gels and creams. For the preparation of cosmetics containing thermal water, it is necessary to study the chemical composition of the water, as it may contain traces of some elements that can be a differentiating element. An example is the presence of iron in thermal water, as this can change the final cosmetic product, as well as the high mineralization of certain waters, with it sometimes being necessary to reduce the amount of total water in the formulation. It is also important to take into account microbial content, since from the point of view of cosmetic preparation, it is not favorable (despite being of interest for certain skin problems). It is necessary to use a system that, without modifying the chemical composition of the thermal water, reduces the microbial load to acceptable levels for the cosmetic, e.g., membrane filtration or by adding preservatives [2].

## 6. Conclusions

Knowledge about thermal waters and their bioactivities has evolved over time, and the various scientific studies designed in the field of dermatology have been essential to understanding their biological effects. In this sense, there is scientific evidence that the therapeutic properties of thermal spring waters are due to their chemical composition, and in particular, to the presence of certain minerals and trace elements that constitute the individual physicochemical profile of each water.

As mentioned throughout this work, the intervention of this type of water with several specific activities on the skin and the absence of side effects reported in different studies allow them to be used as an adjuvant or in the treatment of various skin conditions, being an active ingredient with potential interest for cosmetic formulations. For this purpose, thermal spring water should always be subjected to in vitro and in vivo studies of their biological effects as well as stability and efficacy studies of the final formulation.

This article compiles all the comprehensive information on the studies that exist on thermal spring waters for cosmetic use.

Having said that, and being aware of the potential of these waters in dermatological application, the future perspectives for the use of thermal water as an active ingredient in cosmetic formulations are promising, and further clinical trials are needed to evaluate the efficacy and safety of cosmetic formulations containing these types of waters both as active ingredients or as excipients. Thermal spring waters can thus play an important role in the cosmetic industry, as well as for the treatment of skin conditions; however, it is still necessary to deepen the methods, as they will be used in cosmetic products in order to improve their skin absorption.

**Author Contributions:** Conceptualization, A.R.T.S.A.; methodology and formal analysis, M.R., M.L.M. and A.R.T.S.A.; writing—original draft preparation, A.C.F.; writing—review and editing, M.R., M.L.M. and A.R.T.S.A.; supervision, A.R.T.S.A. All authors have read and agreed to the published version of the manuscript.

**Funding:** This research received no external funding.

**Institutional Review Board Statement:** Not applicable.

**Informed Consent Statement:** Not applicable.

**Data Availability Statement:** Not applicable.

**Conflicts of Interest:** The authors declare no conflict of interest.

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
