# Peer review of "Thermal Spring Waters as an Active Ingredient in Cosmetic Formulations"

_cosmetics, doi:10.3390/cosmetics10010027_

Round 1

Reviewer 1 Report

page 9

The the somparison  between spraying ATSW and photodynamic therapy needs more explanation. It is an interesting study and such  studies are fei un the literature, in general

Another point that it is reffered in the manuscript , but I think it needs more explanation and examples, is the preservation of such formulationas or sprays. 

Author Response

We thank the reviewer for the pertinent advice and comments. We added more information regarding the study of spraying ATSW and photodynamic therapy. Regarding the preservation of the formulations with thermal spring waters, it could be applied a membrane filtration or adding preservatives. The corrections were performed using the “Track Changes” function of MS Word. 

Reviewer 2 Report

The manuscript "THERMAL SPRING WATERS AS AN ACTIVE INGREDIENT IN COSMETIC FORMULATIONS" presents a useful manuscript on current data on use of thermal spring waters as an active ingredient in cosmetic formulations. It is based on  the analysis of previous research data on this topic, conducted by various authors. I consider the manuscript valuable for current knowledge on this topic and further work/practice in dermatology and similar fields. However, there are some suggestions.

INTRODUCTION Is there any article which compares various thermal spring waters (as an active ingredient) in cosmetic formulations? If not, you may emphasize this data. If yes, you may mention the difference between your article and previous articles (e.g. in the Discussion section).

RESULTS: The results are presented in one valuable table. So, Table 1 is very useful for comparison and presentation of different study results. However, please check for the style of writing in the table and details (they should be consistent in different parts of the table) - e.g. data mentioned in Observed effects and Specific activities are somewhat mixed.

Also, authors mention detailed previous results obtained by various studies. So, in the text, maybe they may shorten some parts, where possible. Also, it is needed to check for possible overlapping with original texts.

There are some typos on some parts:, for example interleukin or IL; words should be written in one uniform style (e.g. some words are sometimes written by upper, sometimes by lower letters).

In DISCUSSION: Limitations and strengths of the study should be mentioned, eg. is there possible incorrect data due to bias of investigators?

Author Response

We thank the reviewer for the positive feedback and constructive remarks. Regarding the INTRODUCTION, there are some previous publications related to this topic like reference 1 (published in proceedings) but that did not so deeply focus on the different studies in which thermal waters are used as ingredients for cosmetic formulations.

Regarding the RESULTS, we revised table 1 to check the style of writing.

We also tried to revise the details of the previous results obtained. Furthermore, we tried to avoid typos like the interleukin (always using an abbreviation after the full name in the first appearance) and other small incorrections throughout the manuscript.

Finally, in DISCUSSION, we think that we tried to be impartial avoiding incorrect data due to bias and also focused on some limitations and strengths of the performed study.

The corrections were performed using the “Track Changes” function of MS Word.